# Gauging Variational Inference

**Sungsoo Ahn**[*]     **Michael Chertkov**[†]     **Jinwoo Shin**[*]

[*]School of Electrical Engineering,
Korea Advanced Institute of Science and Technology, Daejeon, Korea
[†1] Theoretical Division, T-4 & Center for Nonlinear Studies,
Los Alamos National Laboratory, Los Alamos, NM 87545, USA,
[†2]Skolkovo Institute of Science and Technology, 143026 Moscow, Russia
[*]{sungsoo.ahn, jinwoos}@kaist.ac.kr     [†]chertkov@lanl.gov

## Abstract

Computing partition function is the most important statistical inference task arising in applications of Graphical Models (GM). Since it is computationally intractable, approximate methods have been used in practice, where mean-field (MF) and belief propagation (BP) are arguably the most popular and successful approaches of a variational type. In this paper, we propose two new variational schemes, coined Gauged-MF (G-MF) and Gauged-BP (G-BP), improving MF and BP, respectively. Both provide lower bounds for the partition function by utilizing the so-called gauge transformation which modifies factors of GM while keeping the partition function invariant. Moreover, we prove that both G-MF and G-BP are exact for GMs with a single loop of a special structure, even though the bare MF and BP perform badly in this case. Our extensive experiments indeed confirm that the proposed algorithms outperform and generalize MF and BP.

## 1   Introduction

Graphical Models (GM) express factorization of the joint multivariate probability distributions in statistics via a graph of relations between variables. The concept of GM has been developed and/or used successfully in information theory [1, 2], physics [3, 4, 5, 6, 7], artificial intelligence [8], and machine learning [9, 10]. Of many inference problems one can formulate using a GM, computing the partition function (normalization), or equivalently computing marginal probability distributions, is the most important and universal inference task of interest. However, this paradigmatic problem is known to be computationally intractable in general, i.e., it is #P-hard even to approximate [11].

The Markov chain monte carlo (MCMC) [12] is a classical approach addressing the inference task, but it typically suffers from exponentially slow mixing or large variance. Variational inference is an approach stating the inference task as an optimization. Hence, it does not have such issues of MCMC and is often more favorable. The mean-field (MF) [6] and belief propagation (BP) [13] are arguably the most popular algorithms of the variational type. They are distributed, fast and overall very successful in practical applications even though they are heuristics lacking systematic error control. This has motivated researchers to seek for methods with some guarantees, e.g., providing lower bounds [14, 15] and upper bounds [16, 17, 15] for the partition function of GM.

In another line of research, which this paper extends and contributes, the so-called re-parametrizations [18], gauge transformations (GT) [19, 20] and holographic transformations [21, 22] were explored. This class of distinct, but related, transformations consist in modifying a GM by changing factors, associated with elements of the graph, continuously such that the partition function stays the same/invariant.[1] In this paper, we choose to work with GT as the most general one among the three

approaches. Once applied to a GM, it transforms the original partition function, defined as a weighted series/sum over states, to a new one, dependent on the choice of gauges. In particular, a fixed point of BP minimizes the so-called Bethe free energy [26], and it can also be understood as an optimal GT [19, 20, 27, 28]. Moreover, fixing GT in accordance with BP results in the so-called loop series expression for the partition function [19, 20]. In this paper we generalize [19, 20] and explore a more general class of GT: we develop a new gauge-optimization approach which results in 'better' variational inference schemes than MF, BP and other related methods.

**Contribution.** The main contribution of this paper consists in developing two novel variational methods, called Gauged-MF (G-MF) and Gauged-BP (G-BP), providing lower bounds on the partition function of GM. While MF minimizes the (exact) Gibbs free energy under (reduced) product distributions, G-MF does the same task by introducing an additional GT. Due to the the additional degree of freedom in optimization, G-MF improves the lower bound of the partition function provided by MF systematically. Similarly, G-BP generalizes BP, extending interpretation of the latter as an optimization of the Bethe free energy over GT [19, 20, 27, 28], by imposing additional constraints on GT, and thus forcing all the terms in the resulting series for the partition function to remain non-negative. Consequently, G-BP results in a provable lower bound for the partition function, while BP does not (except for log-supermodular models [29]).

We prove that both G-MF and G-BP are exact for GMs defined over single cycle, which we call 'alternating cycle/loop', as well as over line graph. The alternative cycle case is surprising as it represents the simplest 'counter-example' from [30], illustrating failures of MF and BP. For general GMs, we also establish that G-MF is better than, or at least as good as, G-BP. However, we also develop novel error correction schemes for G-BP such that the lower bound of the partition function provided by G-BP is improved systematically/sequentially, eventually outperforming G-MF on the expense of increasing computational complexity. Such error correction scheme has been studied for improving BP by accounting for the loop series consisting of positive and negative terms [31, 32]. According to to our design of G-BP, the corresponding series consists of only non-negative terms, which leads to easier systematic corrections to G-BP.

We also show that the proposed GT-based optimizations can be restated as smooth and unconstrained, thus allowing efficient solutions via algorithms of a gradient descent type or any generic optimization solver, such as IPOPT [33]. We experiment with IPOPT on complete GMs of relatively small size and on large GM (up-to 300 variables) of fixed degree. Our experiments indeed confirm that the newly proposed algorithms outperform and generalize MF and BP. Finally, we remark that all statements of the paper are made within the framework of the so-called Forney-style GMs [34] which is general as it allows interactions beyond pair-wise (i.e., high-order GM) and includes other/alternative GM formulations, based on factor graphs [35].

## 2 Preliminaries

### 2.1 Graphical model

**Factor-graph model.** Given (undirected) bipartite factor graph $G = (\mathcal{X}, \mathcal{F}, \mathcal{E})$, a joint distribution of (binary) random variables $x = [x_v \in \{0, 1\} : v \in \mathcal{X}]$ is called a factor-graph Graphical Model (GM) if it factorizes as follows:

$$p(x) = \frac{1}{Z} \prod_{a \in \mathcal{F}} f_a(x_{\partial a}),$$

where $f_a$ are some non-negative functions called factor functions, $\partial a \subseteq \mathcal{X}$ consists of nodes neighboring factor $a$, and the normalization constant $Z := \sum_{x \in \{0,1\}^{\mathcal{X}}} \prod_{a \in \mathcal{F}} f_a(x_{\partial a})$, is called the partition function. A factor-graph GM is called pair-wise if $|\partial a| \leq 2$ for all $a \in \mathcal{F}$, and high-order otherwise. It is known that approximating the partition function is #P-hard in general [11].

**Forney-style model.** In this paper, we primarily use the Forney-style GM [34] instead of factor-graph GM. Elementary random variables in the Forney-style GM are associated with edges of an undirected graph, $G = (\mathcal{V}, \mathcal{E})$. Then the random vector, $x = [x_{ab} \in \{0, 1\} : \{a, b\} \in \mathcal{E}]$ is realized with the probability distribution

$$p(x) = \frac{1}{Z} \prod_{a \in \mathcal{V}} f_a(x_a), \tag{1}$$

where $x_a$ is associated with set of edges neighboring node $a$, i.e. $x_a = [x_{ab} \; : \; b \in \partial a]$ and $Z := \sum_{x \in \{0,1\}^{\mathcal{E}}} \prod_{a \in V} f_a(x_a)$. As argued in [19, 20], the Forney-style GM constitutes a more universal/compact description of gauge transformations without any restriction of generality: given any factor-graph GM, one can construct an equivalent Forney-style (see the supplementary material).

## 2.2 Mean-field and belief propagation

We now introduce two most popular methods for approximating the partition function: the mean-field and Bethe (i.e., belief propagation) approximation methods. Given any (Forney-style) GM $p(x)$ defined as in (1) and any distribution $q(x)$ over all variables, the *Gibbs free energy* is defined as

$$F_{\text{Gibbs}}(q) := \sum_{x \in \{0,1\}^{\mathcal{E}}} q(x) \log \frac{q(x)}{\prod_{a \in \mathcal{V}} f_a(x_a)}. \tag{2}$$

The partition function is related to the Gibbs free energy according to $-\log Z = \min_q F_{\text{Gibbs}}(q)$, where the optimum is achieved at $q = p$ [35]. This optimization is over all valid probability distributions from the exponentially large space, and obviously intractable.

In the case of the mean-field (MF) approximation, we minimize the Gibbs free energy over a family of tractable probability distributions factorized into the following product: $q(x) = \prod_{\{a,b\} \in \mathcal{E}} q_{ab}(x_{ab})$, where each independent $q_{ab}(x_{ab})$ is a proper probability distribution, behaving as a (mean-field) proxy to the marginal of $q(x)$ over $x_{ab}$. By construction, the MF approximation provides a lower bound for $\log Z$. In the case of the Bethe approximation, the so-called *Bethe free energy* approximates the Gibbs free energy [36]:

$$F_{\text{Bethe}}(b) = \sum_{a \in \mathcal{V}} \sum_{x_a \in \{0,1\}^{\partial a}} b_a(x_a) \log \frac{b_a(x_a)}{f_a(x_a)} - \sum_{\{a,b\} \in \mathcal{E}} \sum_{x_{ab} \in \{0,1\}} b_{ab}(x_{ab}) \log b_{ab}(x_{ab}), \tag{3}$$

where *beliefs* $b = [b_a, b_{ab} \; : \; a \in \mathcal{V}, \{a,b\} \in \mathcal{E}]$ should satisfy following 'consistency' constraints:

$$0 \leq b_a, b_{ab} \leq 1, \qquad \sum_{x_{ab} \in \{0,1\}} b_a(x_{ab}) = 1, \qquad \sum_{x'_a \setminus x_{ab} \in \{0,1\}^{\partial a}} b(x'_a) = b(x_{ab}) \quad \forall \{a,b\} \in \mathcal{E}.$$

Here, $x'_a \setminus x_{ab}$ denotes a vector with $x'_{ab} = x_{ab}$ fixed and $\min_b F_{\text{Bethe}}(b)$ is the Bethe estimation for $-\log Z$. The popular belief propagation (BP) distributed heuristics solves the optimization iteratively [36]. The Bethe approximation is exact over trees, i.e., $-\log Z = \min_b F_{\text{Bethe}}(b)$. However, in the case of a general loopy graph, the BP estimation lacks approximation guarantees. It is known, however, that the result of BP-optimization lower bounds the log-partition function, $\log Z$, if the factors are log-supermodular [29].

## 2.3 Gauge transformation

Gauge transformation (GT) [19, 20] is a family of linear transformations of the factor functions in (1) which leaves the the partition function $Z$ invariant. It is defined with respect to the following set of invertible $2 \times 2$ matrices $G_{ab}$ for $\{a,b\} \in \mathcal{E}$, coined *gauges*:

$$G_{ab} = \left[ \begin{array}{cc} G_{ab}(0,0) & G_{ab}(0,1) \\ G_{ab}(1,0) & G_{ab}(1,1) \end{array} \right].$$

The GM, gauge transformed with respect to $\mathcal{G} = [G_{ab}, G_{ba} \; : \; \{a,b\} \in \mathcal{E}]$, consists of factors expressed as:

$$f_{a,\mathcal{G}}(x_a) = \sum_{x'_a \in \{0,1\}^{\partial a}} f_a(x'_a) \prod_{b \in \partial a} G_{ab}(x_{ab}, x'_{ab}).$$

Here one treats independent $x_{ab}$ and $x_{ba}$ equivalently for notational convenience, and $\{G_{ab}, G_{ba}\}$ is a conjugated pair of distinct matrices satisfying the gauge constraint $G_{ab}^{\top} G_{ba} = \mathbb{I}$, where $\mathbb{I}$ is the identity matrix. Then, one can prove invariance of the partition function under the transformation:

$$Z = \sum_{x \in \{0,1\}^{|\mathcal{E}|}} \prod_{a \in \mathcal{V}} f_a(x_a) = \sum_{x \in \{0,1\}^{|\mathcal{E}|}} \prod_{a \in \mathcal{V}} f_{a,\mathcal{G}}(x_a). \tag{4}$$

Consequently, GT results in the gauge transformed distribution $p_\mathcal{G}(x) = \frac{1}{Z} \prod_{a \in \mathcal{V}} f_{a,\mathcal{G}}(x_a)$. Note that some components of $p_\mathcal{G}(x)$ can be negative, in which case it is not a valid distribution.

We remark that the Bethe/BP approximation can be interpreted as a specific choice of GT [19, 20]. Indeed any fixed point of BP corresponds to a special set of gauges making an arbitrarily picked configuration/state $x$ to be least sensitive to the local variation of the gauge. Formally, the following non-convex optimization is known to be equivalent to the Bethe approximation:

$$\begin{aligned} \underset{\mathcal{G}}{\text{maximize}} \quad & \sum_{a \in \mathcal{V}} \log f_{a,\mathcal{G}}(0, 0, \dots) \\ \text{subject to} \quad & G_{ab}^\top G_{ba} = \mathbb{I}, \quad \forall \{a, b\} \in \mathcal{E}, \end{aligned} \quad (5)$$

and the set of BP-gauges correspond to stationary points of (5), having the objective as the respective Bethe free energy, i.e., $\sum_{a \in \mathcal{V}} \log f_{a,\mathcal{G}}(0, 0, \dots) = -F_{\text{Bethe}}$.

# 3 Gauge optimization for approximating partition functions

Now we are ready to describe two novel gauge optimization schemes (different from (5)) providing guaranteed lower bound approximations for $\log Z$. Our first GT scheme, coined Gauged-MF (G-MF), shall be considered as modifying and improving the MF approximation, while our second GT scheme, coined Gauged-BP (G-BP), modifies and improves the Bethe approximation in a way that it now provides a provable lower bound for $\log Z$, while the bare BP does not have such guarantees. The G-BP scheme also allows further improvement (in terms of the output quality) on the expense of making underlying algorithm/computation more complex.

## 3.1 Gauged mean-field

We first propose the following optimization inspired by, and also improving, the MF approximation:

$$\begin{aligned} \underset{q,\mathcal{G}}{\text{maximize}} \quad & \sum_{a \in \mathcal{V}} \sum_{x_a \in \{0,1\}^{\partial a}} q_a(x_a) \log f_{a,\mathcal{G}}(x_a) - \sum_{\{a,b\} \in \mathcal{E}} \sum_{x_{ab} \in \{0,1\}} q_{ab}(x_{ab}) \log q_{ab}(x_{ab}) \\ \text{subject to} \quad & G_{ab}^\top G_{ba} = \mathbb{I}, \quad \forall \{a, b\} \in \mathcal{E}, \\ & f_{a,\mathcal{G}}(x_a) \geq 0, \quad \forall a \in \mathcal{V}, \, \forall x_a \in \{0, 1\}^{\partial a}, \\ & q(x) = \prod_{\{a,b\} \in \mathcal{E}} q_{ab}(x_{ab}), \quad q_a(x_a) = \prod_{b \in \partial a} q_{ab}(x_{ab}), \quad \forall a \in \mathcal{V}. \end{aligned} \quad (6)$$

Recall that the MF approximation optimizes the Gibbs free energy with respect to $q$ given the original GM, i.e. factors. On the other hand, (6) jointly optimizes it over $q$ and $\mathcal{G}$. Since the partition function of the gauge transformed GM is equal to that of the original GM, (6) also outputs a lower bound on the (original) partition function, and always outperforms MF due to the additional degree of freedom in $\mathcal{G}$. The non-negative constraints $f_{a,\mathcal{G}}(x_a) \geq 0$ for each factor enforce that the gauge transformed GM results in a valid probability distribution (all components are non-negative).

To solve (6), we propose a strategy, alternating between two optimizations, formally stated in Algorithm 1. The alternation is between updating $q$, within Step A, and updating $\mathcal{G}$, within Step C. The optimization in Step A is simple as one can apply any solver of the mean-field approximation. On the other hand, Step C requires a new solver and, at the first glance, looks complicated due to nonlinear constraints. However, the constraints can actually be eliminated. Indeed, one observes that the non-negative constraint $f_{a,\mathcal{G}}(x_a) \geq 0$ is redundant, because each term $q(x_a) \log f_{a,\mathcal{G}}(x_a)$ in the optimization objective already prevents factors from getting close to zero, thus keeping them positive. Equivalently, once current $\mathcal{G}$ satisfies the non-negative constraints, the objective, $q(x_a) \log f_{a,\mathcal{G}}(x_a)$, acts as a log-barrier forcing the constraints to be satisfied at the next step within an iterative optimization procedure. Furthermore, the gauge constraint, $G_{ab}^\top G_{ba} = \mathbb{I}$, can also be removed simply expressing one (of the two) gauge via another, e.g., $G_{ba}$ via $(G_{ab}^\top)^{-1}$. Then, Step C can be resolved by any unconstrained iterative optimization method of a gradient descent type. Next, the additional (intermediate) procedure Step B was considered to handle extreme cases when for some $\{a, b\}$, $q_{ab}(x_{ab}) = 0$ at the optimum. We resolve the singularity perturbing the distribution by setting zero probabilities to a small value, $q_{ab}(x_{ab}) = \delta$ where $\delta > 0$ is sufficiently small. In

---
**Algorithm 1** Gauged mean-field
---
1: **Input:** GM defined over graph $G = (\mathcal{V}, \mathcal{E})$ with factors $\{f_a\}_{a \in \mathcal{V}}$. A sequence of decreasing barrier terms $\delta_1 > \delta_2 > \cdots > \delta_T > 0$ (to handle extreme cases).

---
2: **for** $t = 1, 2, \cdots, T$ **do**
3:     **Step A.** Update $q$ by solving the mean-field approximation, i.e., solve the following optimization:

$$\underset{q}{\text{maximize}} \quad \sum_{a \in \mathcal{V}} \sum_{x_a \in \{0,1\}^{\partial a}} q_a(x_a) \log f_{a,\mathcal{G}}(x_a) - \sum_{\{a,b\} \in \mathcal{E}} \sum_{x_{ab} \in \{0,1\}} q_{ab}(x_{ab}) \log q_{ab}(x_{ab})$$

$$\text{subject to} \quad q(x) = \prod_{\{a,b\} \in \mathcal{E}} q_{ab}(x_{ab}), \quad q_a(x_a) = \prod_{b \in \partial a} q_{ab}(x_{ab}), \quad \forall a \in \mathcal{V}.$$

4:     **Step B.** For factors with zero values, i.e. $q_{ab}(x_{ab}) = 0$, make perturbation by setting

$$q_{ab}(x'_{ab}) = \begin{cases} \delta_t & \text{if } x'_{ab} = x_{ab} \\ 1 - \delta_t & \text{otherwise.} \end{cases}$$

5:     **Step C.** Update $\mathcal{G}$ by solving the following optimization:

$$\underset{\mathcal{G}}{\text{maximize}} \quad \sum_{a \in V} \sum_{x \in \{0,1\}^{\mathcal{E}}} q(x) \log \prod_{a \in V} f_{a,\mathcal{G}}(x_a)$$

$$\text{subject to} \quad G_{ab}^\top G_{ba} = \mathbb{I}, \quad \forall \{a, b\} \in \mathcal{E}.$$

---
6: **end for**
---
7: **Output:** Set of gauges $\mathcal{G}$ and product distribution $q$.
---

summary, it is straightforward to check that the Algorithm 1 converges to a local optimum of (6), similar to some other solvers developed for the mean-field and Bethe approximations.

We also provide an important class of GMs where the Algorithm 1 provably outperforms both the MF and BP (Bethe) approximations. Specifically, we prove that the optimization (6) is exact in the case when the graph is a line (which is a special case of a tree) and, somewhat surprisingly, a single loop/cycle with odd number of factors represented by negative definite matrices. In fact, the latter case is the so-called 'alternating cycle' example which was introduced in [30] as the simplest loopy example where the MF and BP approximations perform quite badly. Formally, we state the following theorem whose proof is given in the supplementary material.

**Theorem 1.** *For GM defined on any line graph or alternating cycle, the optimal objective of* (6) *is equal to the exact log partition function, i.e.,* $\log Z$.

### 3.2 Gauged belief propagation

We start discussion of the G-BP scheme by noticing that, according to [37], the G-MF gauge optimization (6) can be reduced to the BP/Bethe gauge optimization (5) by eliminating the non-negative constraint $f_{a,\mathcal{G}}(x_a) \geq 0$ for each factor and replacing the product distribution $q(x)$ by:

$$q(x) = \begin{cases} 1 & \text{if } x = (0, 0, \cdots), \\ 0 & \text{otherwise.} \end{cases} \tag{7}$$

Motivated by this observation, we propose the following G-BP optimization:

$$\underset{\mathcal{G}}{\text{maximize}} \quad \sum_{a \in V} \log f_{a,\mathcal{G}}(0, 0, \cdots)$$

$$\text{subject to} \quad G_{ab}^\top G_{ba} = \mathbb{I}, \quad \forall (a, b) \in \mathcal{E},$$

$$f_{a,\mathcal{G}}(x_a) \geq 0, \quad \forall a \in \mathcal{V}, \forall x_a \in \{0, 1\}^{\partial a}. \tag{8}$$

The only difference between (5) and (8) is addition of the non-negative constraints for factors in (8). Hence, (8) outputs a lower bound on the partition function, while (5) can be larger or smaller then $\log Z$. It is also easy to verify that (8) (for G-BP) is equivalent to (6) (for G-MF) with $q$ fixed to (7). Hence, we propose the algorithmic procedure for solving (8), formally described in Algorithm 2, and it should be viewed as a modification of Algorithm 1 with $q$ replaced by (7) in Step A, also with a properly chosen log-barrier term in Step C. As we discussed for Algorithm 1, it is straightforward to verify that Algorithm 2 also converges to a local optimum of (8) and one can replace $G_{ba}$ by $(G_{ab}^\top)^{-1}$ for each pair of the conjugated matrices in order to build a convergent gradient descent algorithmic implementation for the optimization.

---

**Algorithm 2** Gauged belief propagation

---

1: **Input:** GM defined over graph $G = (\mathcal{V}, \mathcal{E})$ with and factors $\{f_a\}_{a \in \mathcal{V}}$. A sequence of decreasing barrier terms $\delta_1 > \delta_2 > \cdots > \delta_T > 0$.

---

2: **for** $t = 1, 2, \cdots$ **do**
3:   Update $\mathcal{G}$ by solving the following optimization:

$$\underset{\mathcal{G}}{\text{maximize}} \quad \sum_{a \in V} \log f_{a,\mathcal{G}}(0, 0, \cdots) + \delta_t \sum_{a \in V} \sum_{x \in \{0,1\}^{\mathcal{E}}} q(x) \log \prod_{a \in V} f_{a,\mathcal{G}}(x_a)$$

$$\text{subject to} \quad G_{ab}^\top G_{ba} = \mathbb{I}, \quad \forall \{a, b\} \in \mathcal{E}.$$

---

4: **end for**

---

5: **Output:** Set of gauges $\mathcal{G}$.

---

Since fixing $q(x)$ eliminates the degree of freedom in (6), G-BP should perform worse than G-MF, i.e., (8) $\leq$ (6). However, G-BP is still meaningful due to the following reasons. First, Theorem 1 still holds for (8), i.e., the optimal $q$ of (6) is achieved at (7) for any line graph or alternating cycle (see the proof of the Theorem 1 in the supplementary material). More importantly, G-BP can be corrected systematically. At a high level, the "error-correction" strategy consists in correcting the approximation error of (8) sequentially while maintaining the desired lower bounding guarantee. The key idea here is to decompose the error of (8) into partition functions of multiple GMs, and then repeatedly lower bound each partition function. Formally, we fix an arbitrary ordering of edges $e_1, \cdots e_{|\mathcal{E}|}$ and define the corresponding GM for each $e_i$ as follows: $p(x) = \frac{1}{Z_i} \prod_{a \in \mathcal{V}} f_{a,\mathcal{G}}(x_a)$ for $x \in \mathcal{X}_i$, where $Z_i := \sum_{x \in \mathcal{X}_i} \prod_{a \in \mathcal{V}} f_{a,\mathcal{G}}(x)$ and

$$\mathcal{X}_i := \{x \ : \ x_{e_i} = 1, x_{e_j} = 0, x_{e_k} \in \{0, 1\} \quad \forall j, k, \text{ such that } 1 \leq j < i < k \leq |\mathcal{E}|\}.$$

Namely, we consider GMs from sequential conditioning of $x_{e_1}, \cdots, x_{e_i}$ in the gauge transformed GM. Next, recall that (8) maximizes and outputs a single configuration $\prod_a f_{a,\mathcal{G}}(0, 0, \cdots)$. Then, since $\mathcal{X}_i \bigcap \mathcal{X}_j = \emptyset$ and $\bigcup_{i=1}^{|\mathcal{E}|} \mathcal{X}_i = \{0, 1\}^{\mathcal{E}} \backslash (0, 0, \cdots)$, the error of (8) can be decomposed as follows:

$$Z - \prod_a f_{a,\mathcal{G}}(0, 0, \cdots) = \sum_{i=1}^{|\mathcal{E}|} \sum_{x \in \mathcal{X}_i} \prod_{a \in \mathcal{V}} f_{a,\mathcal{G}}(x) = \sum_{i=1}^{|\mathcal{E}|} Z_i, \tag{9}$$

Now, one can run G-MF, G-BP or any other methods (e.g., MF) again to obtain a lower bound $\widehat{Z}_i$ of $Z_i$ for all $i$ and then output $\prod_{a \in \mathcal{V}} f_{a,\mathcal{G}}(0, 0, \cdots) + \sum_{i=1}^{|\mathcal{E}|} \widehat{Z}_i$. However, such additional runs of optimization inevitably increase the overall complexity. Instead, one can also pick a single term $\prod_a f_{a,\mathcal{G}}(x_a^{(i)})$ for $x^{(i)} = [x_{e_i} = 1, x_{e_j} = 0, \ \forall j \neq i]$ from $\mathcal{X}_i$, as a choice of $\widehat{Z}_i$ just after solving (8) initially, and output

$$\prod_{a \in \mathcal{V}} f_{a,\mathcal{G}}(0, 0, \cdots) + \sum_{i=1}^{|\mathcal{E}|} f_{a,\mathcal{G}}(x_a^{(i)}), \qquad x^{(i)} = [x_{e_i} = 1, x_{e_j} = 0, \ \forall j \neq i], \tag{10}$$

as a better lower bound for $\log Z$ than $\prod_{a \in \mathcal{V}} f_{a,\mathcal{G}}(0, 0, \cdots)$. This choice is based on the intuition that configurations partially different from $(0, 0, \cdots)$ may be significant too as they share most of the same factor values with the zero configuration maximized in (8). In fact, one can even choose more configurations (partially different from $(0, 0, \cdots)$) by paying more complexity, which is always

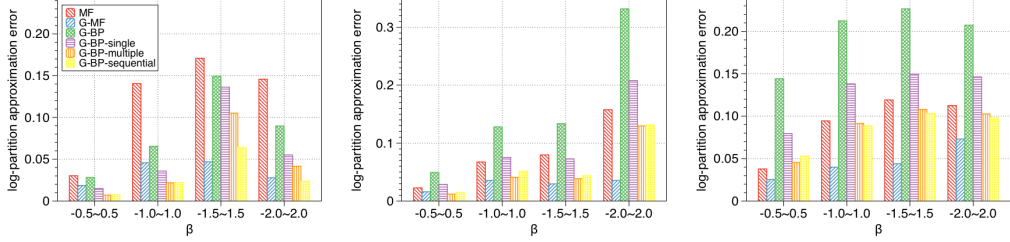

Figure 1: Averaged log-partition approximation error vs interaction strength $\beta$ in the case of generic (non-log-supermodular) GMs on complete graphs of size 4, 5 and 6 (left, middle, right), where the average is taken over 20 random models.

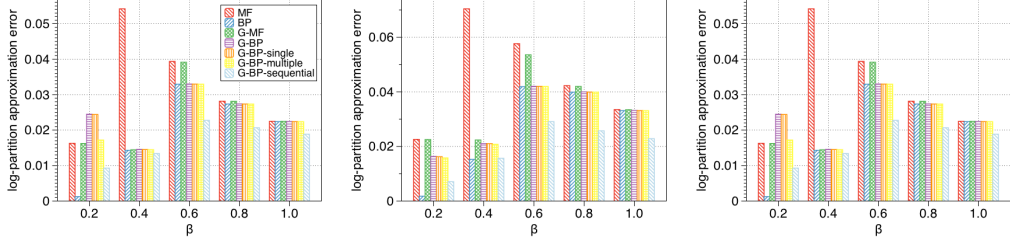

Figure 2: Averaged log-partition approximation error vs interaction strength $\beta$ in the case of log-supermodular GMs on complete graphs of size 4, 5 and 6 (left, middle, right), where the average is taken over 20 random models.

better as it brings the approximation closer to the true partition function. In our experiments, we consider additional configurations $\{x \; : \; [x_{e_i} = 1, x_{e_{i'}} = 1, x_{e_j} = 0, \; \forall \, i, i' \neq j] \text{ for } i' = i, \cdots |\mathcal{E}|\}$, i.e., output

$$\prod_{a \in \mathcal{V}} f_{a,\mathcal{G}}(0,0,\cdots) + \sum_{i=1}^{|\mathcal{E}|} \sum_{i'=i}^{|\mathcal{E}|} f_{a,\mathcal{G}}(x_a^{(i,i')}), \quad x^{(i,i')} = [x_{e_i} = 1, x_{e_{i'}} = 1, x_{e_j} = 0, \; \forall \, j \neq i, i'], \tag{11}$$

as a better lower bound of $\log Z$ than (10).

## 4   Experimental results

We report results of our experiments with G-MF and G-BP introduced in Section 3. We also experiment here with improved G-BPs correcting errors by accounting for single (10) and multiple (11) terms, as well as correcting G-BP by applying it (again) sequentially to each residual partition function $Z_i$. The error decreases, while the evaluation complexity increases, as we move from G-BP-single to G-BP-multiple and then to G-BP-sequential. To solve the proposed gauge optimizations, e.g., **Step C.** of Algorithm 1, we use the generic optimization solver IPOPT [33]. Even though the gauge optimizations can be formulated as unconstrained optimizations, IPOPT runs faster on the original constrained versions in our experiments.[2] However, the unconstrained formulations has a strong future potential for developing fast gradient descent algorithms. We generate random GMs with factors dependent on the 'interaction strength' parameters $\{\beta_a\}_{a \in \mathcal{V}}$ (akin inverse temperature) according to:

$$f_a(x_a) = \exp(-\beta_a |h_0(x_a) - h_1(x_a)|),$$

where $h_0$ and $h_1$ count numbers of 0 and 1 contributions in $x_a$, respectively. Intuitively, we expect that as $|\beta_a|$ increases, it becomes more difficult to approximate the partition function. See the supplementary material for additional information on how we generate the random models.

In the first set of experiments, we consider relatively small, complete graphs with two types of factors: random generic (non-log-supermodular) factors and log-supermodular (positive/ferromagnetic) factors. Recall that the bare BP also provides a lower bound in the log-supermodular case [29], thus making the comparison between each proposed algorithm and BP informative. We use the log partition approximation error defined as $|\log Z - \log Z_{\text{LB}}|/|\log Z|$, where $Z_{\text{LB}}$ is the algorithm

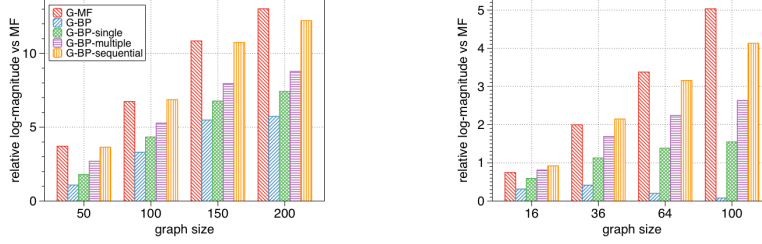

Figure 3: Averaged ratio of the log partition function compared to MF vs graph size (i.e., number of factors) in the case of generic (non-log-supermodular) GMs on 3-regular graphs (left) and grid graphs (right), where the average is taken over 20 random models.

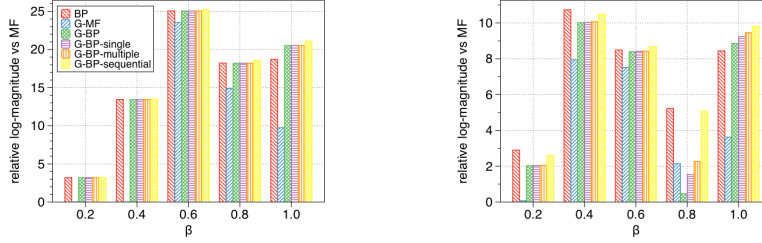

Figure 4: Averaged ratio of the log partition function compared to MF vs interaction strength $\beta$ in the case of log-supermodular GMs on 3-regular graphs of size 200 (left) and grid graphs of size 100 (right), where the average is taken over 20 random models.

output (a lower bound of $Z$), to quantify the algorithm's performance. In the first set of experiments, we deal with relatively small graphs and the explicit computation of $Z$ (i.e., the approximation error) is feasible. The results for experiments over the small graphs are illustrated in Figure 1 and Figure 2 for the non-log-supermodular and log-supermodular cases, respectively. Figure 1 shows that, as expected, G-MF always outperforms MF. Moreover, we observe that G-MF typically provides the tightest low-bound, unless it is outperformed by G-BP-multiple or G-BP-sequential. We remark that BP is not shown in Figure 1, because in this non-log-supermodular case, it does not provide a lower bound in general. According to Figure 2, showing the log-supermodular case, both G-MF and G-BP outperform MF, while G-BP-sequential outperforms all other algorithms. Notice that G-BP performs rather similar to BP in the log-supermodular case, thus suggesting that the constraints, distinguishing (8) from (5), are very mildly violated.

In the second set of experiments, we consider more sparse, larger graphs of two types: 3-regular and grid graphs with size up to 200 factors/300 variables. As in the first set of experiments, the same non-log-supermodular/log-supermodular factors are considered. Since computing the exact approximation error is not feasible for the large graphs, we instead measure here the ratio of estimation by the proposed algorithm to that of MF, i.e., $\log(Z_{\mathrm{LB}}/Z_{\mathrm{MF}})$ where $Z_{\mathrm{MF}}$ is the output of MF. Note that a larger value of the ratio indicates better performance. The results are reported in Figure 3 and Figure 4 for the non-log-supermodular and log-supermodular cases, respectively. In Figure 3, we observe that G-MF and G-BP-sequential outperform MF significantly, e.g., up-to $e^{14}$ times better in 3-regular graphs of size 200. We also observe that even the bare G-BP outperforms MF. In Figure 4, algorithms associated with G-BP outperform G-MF and MF (up to $e^{25}$ times). This is because the choice of $q(x)$ for G-BP is favored by log-supermodular models, i.e., most of configurations are concentrated around $(0, 0, \cdots)$ similar to the choice (7) of $q(x)$ for G-BP. One observes here (again) that performance of G-BP in this log-supermodular case is almost on par with BP. This implies that G-BP generalizes BP well: the former provides a lower bound of $Z$ for any GMs, while the latter does only for log-supermodular GMs.

## 5 Conclusion

We explore the freedom in gauge transformations of GM and develop novel variational inference methods which result in significant improvement of the partition function estimation. We note that the GT methodology, applied here to improve MF and BP, can also be used to improve and extend utility of other variational methods.

**Acknowledgments**

This work was supported in part by the National Research Council of Science & Technology (NST) grant by the Korea government (MSIP) (No. CRC-15-05-ETRI), Institute for Information & communications Technology Promotion(IITP) grant funded by the Korea government(MSIT) (No.2017-0-01778, Development of Explainable Human-level Deep Machine Learning Inference Framework) and ICT R&D program of MSIP/IITP [2016-0-00563, Research on Adaptive Machine Learning Technology Development for Intelligent Autonomous Digital Companion].

## Footnotes

[1]See [23, 24, 25] for discussions of relations between the aforementioned techniques.

[2] The running time of the implemented algorithms are reported in the supplementary material.

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
