[Supplementary Material]

# A Construction of Forney-style model equivalent to factor-graph model

In this Section, we describe construction of a Forney-style GM equivalent to the factor-graph GM. Consider a factor-graph GM defined on graph $G = (\mathcal{X}, \mathcal{F}, \mathcal{E})$ with factors $\{f_a\}_{a \in \mathcal{F}}$. Then one introduces the following Forney-style GM defined over the graph $(\mathcal{V}, \mathcal{E})$ with factors $\{f_a^\dagger\}_{a \in \mathcal{V}}$

$$\mathcal{V} \leftarrow \mathcal{X} \cup \mathcal{F}, \qquad f_a^\dagger \leftarrow f_a, \quad \forall a \in \mathcal{F},$$

$$f_a^\dagger(x_a) \leftarrow \begin{cases} 1 & \text{if } x_a = (1, 1, \cdots) \text{ or } (0, 0, \cdots) \\ 0 & \text{otherwise} \end{cases} \quad , \forall a \in \mathcal{X}.$$

One observes that if the factor-graph GM (possibly, of high-order) is sparse, i.e., the maximum degree of $(\mathcal{X}, \mathcal{F}, \mathcal{E})$ is small, then the equivalent Forney-style GM is too. See Figure 5 for illustration.

Figure 5: Example of the transformation from the factor-graph GM (left) to the Forney-style GM (right). Factors denoted as '=' constrains adjoining variables to have the same value. Originally, the factor-graph GM had 3 variables $(x_1, x_2, x_3)$ and 2 factors $(a, b)$. In the equivalent Forney-style GM, there are 6 variables $(x_{1a}, x_{1b}, x_{2a}, x_{2b}, x_{3a}, x_{3b})$ and 5 factors ($a, b$ and three '=' factors).

# B Proof of Theorem 1

To prove Theorem 1 one, first, shows that the line graph GM can be gauge transformed into a distribution equivalent to the alternating cycle GM. Then it is sufficient for proving Theorem 1 to consider only the case of an alternating cycle.

Consider a GM defined on a line graph $G = (\mathcal{V}, \mathcal{E})$ with $\mathcal{V} = \{a_1, a_2, \cdots, a_n\}$ and edges $\mathcal{E} = \{\{a_1, a_2\}, \{a_2, a_3\}, \cdots, \{a_{n-1}, a_n\}\}$. Then the gauge transformed factor $f_{a_i, \mathcal{G}}$ can be expressed as:

$$f_{a_i, \mathcal{G}} = G_{a_i a_{i-1}}^\top f_{a_i} G_{a_i a_{i+1}},$$

where we used the fact that the size/cardinality of the factor is 2. Next, we 'flip' factor $f_2$, associated with the node number 2, such that there exist an odd number of negative definite factors among $f_2, \cdots f_{n-1}$, i.e., the flipping sets

$$G_{a_1 a_2}, G_{a_2 a_1} = \begin{bmatrix} 0 & 1 \\ 1 & 0 \end{bmatrix}, \tag{12}$$

thus resulting in reversing the sign of $\det(f_{a_2})$. If $f_{a_2}$ is non-invertible, i.e. $\det(f_{a_2}) = 0$, we instead flip $f_3$ and so on. If all factors are non-invertible, the resulting distribution is a product distribution and one can easily find the optimal $q$ for the corresponding line graph, which completes the proof. Otherwise, we 'join' the endpoints $a_1, a_n$ into $a_0$ by introducing a non-invertible factor $f_0 = f_1 f_n^\top$, which results in an alternating cycle with the probability distribution identical to the one of a line graph GM.

Our next step is to prove Theorem 1 for an alternating cycle GM. Our high level logic here is as follows. We first fix the distribution $q$ of (6) according to

$$q(x) = \begin{cases} 1 & \text{if } x = (0, 0, \cdots), \\ 0 & \text{otherwise.} \end{cases},$$

and then show that the GM can be gauge transformed into a distribution with a nonzero probability concentrated only at $(0, 0, \cdots)$. The resulting objective of (6) will become exactly the partition function. To implement this logic, consider an alternating cycle defined on some graph $G =$

$(\mathcal{V}, \mathcal{E})$ with $\mathcal{V} = \{a_1, a_2, \cdots, a_n\}$ and edges $\mathcal{E} = \{\{a_1, a_2\}, \{a_2, a_3\}, \cdots, \{a_{n-1}, a_n\}, \{a_n, a_1\}\}$. Observe that, that the gauge transformed factor, $\prod_i f_{a_i, \mathcal{G}}$, and the original factor, $\prod_i f_{a_i}$, share a pair of eigenvalues $\lambda_1, \lambda_2$ due to the following relationship:

$$\prod_i f_{a_i, \mathcal{G}} = G_{a_n a_1}^{-1} \prod_i f_i G_{a_n a_1}$$

One finds that $\lambda_1 \lambda_2 = \prod \det(f_i) \leq 0$ since there exist an odd number of negative definite factors in the cycle. Moreover, $\lambda_1 + \lambda_2 > 0$ because the diagonal sum, $\prod_i f_i$, is equivalent to the partition function of GM. Thus one can assume, without loss of generality, that $\lambda_1 > 0$ and $\lambda_2 < 0$.

Next, utilizing a simple linear algebra, one derives

$$Q_2^{-1} Q_1 G_{a_n a_1} \prod_i f_{i, \mathcal{G}} Q_1^{-1} Q_2 = \begin{bmatrix} \lambda_1 + \lambda_2 & \lambda_1 \\ -\lambda_2 & 0 \end{bmatrix},$$

where $Q_1$ and $Q_2$ are matrices whose $j$-th column is an eigen-vector of $\prod_i f_i$ and, $\begin{bmatrix} \lambda_1 + \lambda_2 & \lambda_1 \\ -\lambda_2 & 0 \end{bmatrix}$, respectively. Now let

$$G_{a_n a_1} = Q_1^{-1} Q_2, \qquad G_{a_{i-1} a_i} = (f_i G_{a_i a_{i+1}}^\top)^{-1} \quad \text{for} \quad i = 2, \cdots n,$$

where $a_{n+1} = a_1$. Here we assume that there exists at most one non-invertible factor in the GM and $f_2, \cdots, f_n$ are invertible so that $(f_i G_{a_i a_{i+1}}^\top)^{-1}$ is defined properly. Otherwise, the GM can be decomposed into separate line graphs and the proof can be applied recursively. Then the gauge transformed factors become:

$$f_{a_1, \mathcal{G}} = \begin{bmatrix} \lambda_1 + \lambda_2 & \lambda_1 \\ -\lambda_2 & 0 \end{bmatrix}, \quad f_{a_i, \mathcal{G}} = \begin{bmatrix} 1 & 0 \\ 0 & 1 \end{bmatrix} \quad \forall i \neq 1,$$

which corresponds to a GM with objective of (6) to be equal to the log partition function. This completes the proof of the Theorem 1.

## C   Generating GM instances (for experiments)

In this section, we provide more details on our experimental setups reported in in Section 4. First, we explain how the two types of factors, non-log-supermodular and log-supermodular, were constructed. In the generic case (of non-log-supermodular factors), i.e., correspondent to Figure 1 and Figure 3, one generates factor by first drawing the interaction strength vector at random from the i.i.d. uniform distribution over the interval $[-T, T]$ for some $T > 0$, i.e., $\beta_a \sim \mathcal{U}(-T, T)$. Then, in order to introduce a bias, we add an external variable $y_a$, i.e., half-edge, as follows:

$$f_a(x_a) = \exp(\beta_a |h_0(x_a \cup y_a) - h_1(x_a \cup y_a)|),$$

where $y_a$ is either $\{0\}$ or $\{1\}$ with probability $1/2$ each. More specifically in experiments resulted in Figure 1 one varies $T$ while in the experiments resulted in Figure 3 one fixes $T$ to 1.0, i.e., $\beta_a \sim [-1.0, 1.0]$. Next, in the case of the log-supermodular factors, i.e., setting resulted in Figure 2 and Figure 4, one generates log-supermodular factors by drawing the interaction strength vector from normal distribution with the average $T > 0$ and the variance, $10^{-4}$, i.e., $\beta_a \sim \mathcal{N}(T, 10^{-4})$. Note that there exist no bias in the factors and even though the distribution of the interaction strength is normal, it is highly likely to observe a positive value concentrated around $T$.

## D   Running time comparison of algorithms

In this section, we report experimental results on running time of our proposed algorithms. To this end, we consider 3-regular and grid graphs with varying size with non-log-supermodular factors, i.e., the setting is identical to experiment for Figure 3. Running time was measured by total elapsed time for optimization of intermediate sub-problems, e.g., **Step A.**, **Step C.** of Algorithm 1. Results for G-BP-single and G-BP-multiple were omitted since they solve the same number of optimization as in G-BP. We remark that implementation of algorithms were not fully optimized. Especially, we expect a considerable boost in speed of gauge optimization when generic solver (IPOPT) is replaced by

| graph size | MF | G-MF | G-BP | G-BP-sequential |
|:---:|:---:|:---:|:---:|:---:|
| 50 | 0.11 | 0.45 | 2.30 | 21.33 |
| 100 | 0.14 | 0.97 | 2.72 | 74.54 |
| 150 | 0.25 | 1.62 | 6.19 | 161.77 |
| 200 | 0.31 | 2.38 | 11.31 | 292.90 |

| graph size | MF | G-MF | G-BP | G-BP-sequential |
|:---:|:---:|:---:|:---:|:---:|
| 16 | 0.02 | 0.18 | 26.80 | 29.78 |
| 36 | 0.05 | 0.59 | 69.44 | 87.88 |
| 64 | 0.14 | 1.24 | 106.03 | 174.39 |
| 100 | 0.40 | 2.51 | 182.98 | 366.74 |

Table 1: Running time (in seconds) vs graph size (i.e., number of factors) in generic (non-log-supermodular) GMs on 3-regular (top) and grid (bottom) graphs, where average is taken over 20 random models.

efficient algorithm specifically designed for the purpose. The experiments were performed using a machine with Intel CPU (Intel(R) Xeon(R) CPU E5-2630, 2.20GHz) and 32 GB RAM. The results are reported in Table 1. Overall, observed running time is ranked as: MF < G-MF < G-BP < G-BP-sequential. Remarkably, we observed each gauge optimization in G-MF being solved much faster than that of G-BP, especially in grid graphs. This is because G-BP tries to solve an extreme case of G-MF, where small precision is required. The performance of MF and G-BP-sequential is as expected, since MF is subroutine of G-MF and G-BP is subroutine of G-BP-sequential. Finally, we argue that G-MF and G-BP are scalable to large graph, since increase in running time with respect to increase in graph size are comparable to MF, e.g., MF gets up to 20 times slower in grid graph as its size grows up to 100, while Gauged-MF and Gauged-BP gets slower 13 and 9 times slower respectively.