[Reviews · NeurIPS 2017]

Reviewer 1



The paper presents two new approaches to variational inference for graphical models with binary variables - an important fundamental problem. By adding a search over gauge transformations (which preserve the partition function) to the standard variational search over (pseudo-)marginals, the authors present methods called gauged mean field (G-MF, Alg 1) and gauged belief propagation (G-BP, Alg 2). Very helpfully, both are guaranteed to yield a lower bound on the true partition function - this may be important in practice and enables empirical testing for improved performance even for large, intractable models (since higher lower bounds are better). Initial experiments demonstrate the benefits of the new methods. The authors demonstrate technical skill and good background knowledge. The new approaches seem powerful and potentially very helpful in practice, and appear very sensible and natural when introduced in Sec 3.1 given the earlier background. The discussion and extensions in Sec 3.2 are interesting. Theorem 1 is a strong theoretical result - does it hold in general for a graph with at most one cycle? However, I have some important questions/concerns. If these were fixed, then this could be a strong paper: 1. Algorithm details and runtime. Both algorithms require a sequence of decreasing barrier terms \delta_1 > \delta_2... but I don't see any comments on how these should be set? What should one do in practice and how much impact could this have on runtime and accuracy? There appear to be no comments at all on runtime. Both algorithms will likely take much longer than the usual time for MF or BP. Are we guaranteed an improvement at each iteration? Even if not, because we always have a lower bound, we could use the max so far to yield an improving anytime algorithm. Without any runtime analysis, the empirical accuracy comparisons are not really meaningful. There are existing methods which will provide improved accuracy for more computation, and comparison would be very helpful (e.g. AIS or clamping approaches such as Weller and Jebara NIPS 2014; or Weller and Domke AISTATS 2016, which shows monotonic improvement for MF and TRW when each additional variable is clamped). Note that with clamping methods: if a model has at most one cycle, then by clamping any one variable on the cycle and running BP, we obtain the exact solution (compare to Theorem 1). More generally, if sufficient variables are clamped st the remainder is acyclic then we obtain the exact solution. Of course, for a graph with many cycles this might take a long time - but this shows why considering accuracy vs runtime is important. What is the quality of the marginals returned by the new approaches? Does a better partition function estimate lead to better marginals? 2. Explanation of the gauge transformation. Given the importance here of the gauge transformation, which may be unfamiliar to many readers, it would help significantly if the explanation could be expanded and at least one example provided. I recognize that this is challenging given space constraints - even if it were in the Appendix it would be a big help. l. 177-179 please elaborate on why (8) yields a lower bound on the partition function. Minor details: 1.238-240 Interesting that G-BP performs similarly to BP for log-supermodular models. Any further comment on this? If/when runtime is added, we need more details on which method is used for regular BP. A few typos - l. 53 alternative -> alternating (as just defined in the previous sentence). I suggest that 'frustrated cycle' may be a better name than 'alternating cycle' since a balanced cycle of even length could have edge signs which alternate. l. 92 two most -> two of the most l. 134-5 on the expense of making underlying -> at the expense of making the underlying In Refs, [11] ising -> Ising; [29] and [30] bethe -> Bethe

Reviewer 2



This paper introduces a variant of variational mean field and belief propagation using Gauge transformation. The Gauge transformation that is used was introduced in refs [19,20] (as far as I can tell). The G-MF and G-BP (with variants) algorithms are interesting and provide a better bound on the partition function that the canonical MF. Presented numerical results show considerable improvement in precision for some examples of graphical models. What is not very clear from the paper is whether these approaches are scalable to really large graphical models. The big advantage of both variational mean field and BP is that they can be used in a distributed way on really large datasets. The authors only mention more efficient implementation of the method in the conclusion as a part of future work. The choice of the graphical models used for the numerical experiments could be better motivated. While this work seems to have good theoretical interest, it would be nice to have examples of more practical applications. This is also only mentioned as a subject of future work. In conclusion, the paper is rather interesting theoretically offering a new variant of generic variational algorithm, it is less clear if it offers a perspective for practical applicability due to the computational cost. I've read the author's feedback and took it into account in my score.

Reviewer 3



Summary: The paper proposes two novel variational methods for approximating the partition function of discrete graphical models. Starting from a known connection between gauge transformations and existing variational inference methods, belief propagation (BP) and mean-field (MF), the paper proposes a more general optimization problem with additional degrees of freedom (with respect to existing methods). This allows them to obtain provable lower bounds, and tighter approximations. In particular, they prove that the method is exact for certain (simple) graph structures where BP fails. The method is evaluated experimentally on synthetic models, showing improvements over simple baselines. Comments: The writing is excellent, and the ideas are presented nicely. A few examples for the not-so-common "dual" graph represetation would have been nice, but it's fine given the space limitations. The theoretical analysis is solid and represents a nice contribution to the field. I like how the approach seems to be orthogonal to many other exisitng approaches for improving the accuracy (e.g., structured mean field, etc), and therefore potentially complementary. It's likely that there will be follow up work on this line of work. My only criticism is on the experimental evaluation, which is below standard. The algorithm is compared to simple baselines on synthetic models only, but there is no indication of runtime. There are ways of improving the approximation obtained with plain MF and BP methods. For example, multiple random initializations or clamping variables to estimate partition functions of simpler models (and then sum up). The key question of whether the proposed approach pays off in terms of computational cost is left unanswered. Minor: - It is said that Algorithm 1 (line 160) converges to a local optimum of (6). It would be good to justify this better, given that (block) coordinate descent is not always guaranteed to converge - Notation in the bottom part of page 6 is very confusing - clarify the (0,0,...) notation in (5)